# Neurodiversity Meets Colors: Does Position Awareness Destroy Generalization in Brain Graph Learning?

**Matheo Angelo Pereira Dantas[1], Caterina Graziani[2], Leo Sampaio Ferraz Ribeiro[1],**
**André Carlos Ponce de Leon Ferreira de Carvalho[1]**
[1]Institute of Mathematical and Computer Sciences, University of São Paulo (ICMC-USP)
[2]Department of Information Engineering and Science, University of Siena
`matheoangelo@usp.br`
`caterina.graziani2@unisi.it`
`{leo.ribeiro, andre}@icmc.usp.br`

## Abstract

Graph Neural Networks (GNNs) rely on permutation invariance to exploit symmetries in graph data using principles of Geometric Deep Learning. However, in machine learning models that process fMRI data using a brain atlas, each node corresponds to a region with its own position and neurological function. Thus, permutation invariance would make the model unaware of these aspects, causing a significant loss of biological interpretability and predictive information. For this reason, many GNN architectures opt for assigning each ROI ("Region Of Interest" in the brain) a unique node representation, either explicitly or implicitly through feature engineering, before using the graph as input for the GNN. In this theoretical study, we investigate the consequences of that choice. First, we prove that, if each ROI is explicitly identified with a unique color, it is possible to achieve perfect expressivity using a GNN with a single max-aggregation message-passing layer, which suffices to attain the maximal Rademacher complexity and very loose VC dimension's bounds. Building on that, we derive generalization bounds based on concrete parameters of the model, such as ROI embedding dimension and atlas size, revealing ways in which this tradeoff could manifest in practice. These findings are particularly relevant in the context of fMRI graph learning, where, despite severe struggles with overfitting and data scarcity, generalization theory is still underexplored.

## 1 Introduction and motivation

The brain is a complex organ that dictates the functioning of the cognition and physiology of the body. However, its complexity, coupled with the inherent diversity of human neurology (Seghier & Price, 2018), poses a challenge to the comprehension of neurological patterns that correlate with observable phenotypical differences between individuals, where many conditions, such as neurodevelopmental disabilities (Chapman & Botha, 2023) and mental health disorders (American Psychiatric Association, 2013), have no biomarkers known to science. In the absence of biomarkers, diagnosis is based primarily on qualitative evaluation with specialized psychology and psychiatry professionals (American Psychiatric Association, 2013), in a process that can be economically inaccessible (Durkin et al., 2015) and affected by sociodemographic biases such as gender (Lai et al., 2017) and ethnicity (Mandell et al., 2009). As a consequence, individuals with these conditions often fail to access adaptations that could improve their quality of life, such as legal disability rights (United Nations, 2006), psychotherapy (Chapman & Botha, 2023), and pharmacotherapy (Thase et al., 1997).

To deal with these difficulties, modern neuroscience research investigates the use of machine learning techniques to process neuroimaging data. A particularly relevant line of research explores the use Graph Neural Networks (GNNs) to process Functional Magnetic Resonance Imaging (fMRI) datasets. These models use a brain atlas to replicate functional connectivity as a graph, where each

node represents a "Region Of Interest" (ROI) in the brain mapped by the atlas, and each edge represents a highly positive temporal correlation between the brain activity of the two ROIs (Cui et al., 2022). These models can be used to classify new images as belonging to either the target condition or to the control group. They can also be used to explore connectivity patterns to discover possible neurological biomarkers (Li et al., 2021; Zheng et al., 2024b). GNNs have been successfully used in several previous studies in natural sciences (Choma et al., 2018; Stokes et al., 2020; Wang et al., 2023), thus being natural candidates for analysis in neuroscience. However, the current state-of-the-art in fMRI data still struggles with overfitting, where GNNs frequently cannot perform better than shallow baselines (Xu et al., 2025a; Cui et al., 2022).

Despite the need for a better understanding of the generalization problems faced by GNNs in neuroscience, generalization theory is still underexplored. Although there are several attempts to address overfitting (Tang et al., 2025), they do not provide rigorous theoretical frameworks to understand generalization. Meanwhile, generalization theory is a very active area of research within graph machine learning (Vasileiou et al., 2025), which includes the study of the relation between generalization and expressivity (Xu et al., 2018), whereby GNNs able to express more complex functions of graph connectivity often have weaker generalization guarantees (Carrasco et al., 2025).

The problem of overfitting in GNNs when applied to fMRI can be due to the scarcity of medical datasets (Tang et al., 2025). However, there is an overlooked vulnerability present in many modern models that may amplify the problem: the lack of permutation invariance, enforced through node feature engineering. Classical GNNs use permutation symmetries to exploit the underlying low-dimensional structure of graphs considering their geometry (Bronstein et al., 2021). This approach can remove node identities, as the model is only sensitive to the existing node features and their connectivity structure. Node identity is important in brain network analysis, since each node corresponds to a specific ROI. Thus, the connectivity patterns of the graph are strongly affected by the regions where they occur in the brain, and biomarker extraction also generally aims to highlight particular ROIs and ROI connections that are relevant for the target brain label (Xu et al., 2025a; Li et al., 2021). Due to these reasons, the models currently studied in the literature often use node features to make the model aware of these factors. Examples of such features include using an identity matrix, which basically identifies each ROI with a one-hot vector (Cui et al., 2022; Bessadok et al., 2023), and connection profile, where the corresponding row of each node in the (often weighted) adjacency matrix is used as a feature vector (Cui et al., 2022; Kan et al., 2022). However, making the model position-aware through these features removes an important protection against the curse of dimensionality in GNNs (Bronstein et al., 2021), exacerbating generalization risks.

Another aspect of GNNs that deserves attention in this context is expressivity. Brain network analysis is heavily focused on the connectivity structure of the input graph (Bessadok et al., 2023), so it is desirable to have models that are able to express these complex connectivity patterns; because of that, many models use the Graph Isomorphism Network (GIN (Xu et al., 2018)) architecture, often with the goal of leveraging its well-known expressive power (Kim et al., 2021; Zhang et al., 2024b; Zheng et al., 2024b; Thapaliya et al., 2025; Meng et al., 2025). However, the current literature lacks deep theoretical foundations to understand expressivity in fMRI graphs, so there is little understanding on how an architecture like GIN can improve expressivity or even if it is necessary at all. This is especially important in the face of our need for better generalization, as it is known that higher expressivity comes at the cost of worse generalization guarantees (Carrasco et al., 2025).

**Main contributions.** In this paper, we provide a theoretical analysis of the risks that position awareness can create for generalization in brain network analysis. We show that position awareness allows GNNs to have perfect expressivity with a single max-aggregation message-passing layer and that practically leads to a worst-case Rademacher Complexity and $O(n!)$ VC dimension, where $n$ is the size of the brain atlas. We also relate these results to VC dimension bounds that link the Weisfeiler-Lehman test to the parameters of the model (Morris et al., 2023), such as atlas size or embedding dimension, and study, quantitatively, how this tradeoff can manifest in practice.

The structure of the paper is organized as follows. In Section 2, we introduce the main definitions and concepts that will be used throughout the paper. Section 3 presents the theoretical developments of the study, with novel results on expressivity and generalization bounds. Section 4 addresses the main limitations that demand further research and Section 5 draws the main conclusions from the study. After that, we have an ethics statement, which will be complemented with author positionality

information in the camera-ready version. We also include an Appendix section for more details on related work, theoretical background, proofs, and promising research problems.

## 2  DEFINITIONS AND PRELIMINARIES

### 2.1  GRAPHS

This paper will use the following definitions associated with graphs:

1. **Non-colored graphs:** a non-colored graph is a tuple $G = (V, E)$ where V is its set of vertices or nodes, and $E \subset (V \times V)$ is its set of edges. It is assumed that all graphs are undirected, i. e. $(u, v) \in E \iff (v, u) \in E$, and that there is at most one edge between the same pair of nodes (except for the two directions $(u, v)$ and $(v, u)$ of the edge).

2. **Colored graphs:** a colored graph (which we will also refer to as just "graph") is defined as a tuple $G = (V, E, c)$, where $V$ and $E$ are defined as above and $c : V \to C$ is a function that assigns a color $c(u) \in C$ to each node $u \in V$. We denote by $\mathcal{G}$ the space of finite colored graphs. We may additionally denote colored graphs as $G = (V, E)$ followed by a separate color function $c$.

3. **Isomorphism:** a *structural* isomorphism between two graphs $G = (V, E)$ and $G' = (V', E')$ is a bijection $\phi : V \to V'$ such that $(u, v) \in E \iff (\phi(u), \phi(v)) \in E'$. An isomorphism between the two colored graphs $G$ and $G'$ is a structural isomorphism $\phi$ such that $c(u) = c(\phi(u))$ for all $u \in V$.

4. **Automorphism:** a *structural* automorphism of a colored graph $G = (V, E, c)$ is a structural isomorphism $\pi : V \to V$ between the graph and itself. An automorphism of a colored graph $G$ is a structural automorphism $\pi$ that preserves node colors, i.e. $c(u) = c(\pi(u))$ for all $u \in V$. A (structural) automorphism is a trivial automorphism if $\pi(u) = u, \forall u \in V$. Otherwise, it is called non-trivial[1].

5. **Asymmetric graphs:** a graph is said to be asymmetric if its only structural automorphism is the trivial automorphism $\pi(u) = u$. We call it non-asymmetric otherwise.

### 2.2  GRAPH NEURAL NETWORKS

**Graph Neural Networks (GNNs).** A GNN $\Phi(\mathbf{X}, \mathbf{A})$ receives as input a graph represented by an adjacency matrix $\mathbf{A}$, where $\mathbf{A}_{ij} = 1$ if there is an edge between $i$ and $j$, and $\mathbf{A}_{ij} = 0$ otherwise; and a feature matrix $\mathbf{X}$, where the $i$-th row corresponds to the feature vector $\mathbf{x}_i$ of node $i$. For all node $u$, we define $\mathbf{h}_u^{(0)} = \mathbf{x}_u$ and iteratively update the embedding $\mathbf{h}_u$ with a message-passing layer:

$$\mathbf{h}_u^{(k+1)} = \text{UPDATE}^{(k)}\left(\mathbf{h}_u^{(k)}, \text{AGGREGATE}^{(k)}(\{\{\mathbf{h}_v^{(k)} : v \in \mathcal{N}(u)\}\})\right), \tag{1}$$

where $\mathcal{N}(u)$ is the set of nodes directly connected to $u$. After $L$ layers, it applies a READOUT function to all feature vectors to create a single vector representation of the graph, $\mathbf{h}^{(L)} = \text{READOUT}\left(\{\{\mathbf{h}_v^{(L)} : v \in V\}\}\right)$. The final graph-level prediction is then given by

$$\Phi(\mathbf{X}, \mathbf{A}) = \psi(\mathbf{h}^{(L)})$$

where $\psi$ is a classifier such as an MLP.

**Permutation Invariance.** Standard GNN graph-level classifiers are invariant to permutation, meaning that the following equation holds:

$$\Phi(\mathbf{X}, \mathbf{A}) = \Phi(\mathbf{\Pi X}, \mathbf{\Pi A \Pi}^T), \tag{2}$$

---

[1]If all node colors are unique (that is the case of the graphs in our brain network modeling) the colored graph has only the trivial automorphism. However, the underlying graph structure $G = (V, E)$ may still admit non-trivial structural automorphisms.

where $\mathbf{\Pi}$ is a permutation matrix (each row or column has exactly one entry equal to 1 and all others equal to 0) associated with a permutation $\pi$ in the symmetric group $\mathbb{S}_n$. This matrix formulation corresponds to the action of $\mathbb{S}_n$ on graphs, which we formalize in Appendix B.1.

GNNs enforce invariance by using permutation-invariant AGGREGATE and READOUT functions in Equation 1 such as sum, max, or mean. However, when node features *uniquely identify each node*, any permutation necessarily changes the feature matrix, effectively breaking the symmetries that permutation invariance is meant to exploit.

**The Weisfeiler-Lehman (1-WL) Isomorphism Test.** The 1-WL Isomorphism Test (Weisfeiler & Leman, 1968) is a graph isomorphism test heuristic that is known to directly bound the expressive power of GNNs, where, if 1-WL cannot distinguish two given graphs, a message-passing GNN will always have the same output for both of them (Xu et al., 2018).

Let $G = (V, E)$ and $G' = (V', E')$ be two colored graphs, which we will attempt to classify as being isomorphic to each other. After having each of the initial nodes of both graphs assigned with a color $c_v^{(0)}$, 1-WL iteratively updates each node color through the following function:

$$c_v^{(k+1)} = \text{HASH}\left(c_v^{(k)}, \left\{\!\!\left\{ c_u^{(k)} : u \in \mathcal{N}(v) \right\}\!\!\right\}\right) \tag{3}$$

Where $\{\!\{...\}\!\}$ is a multiset and HASH is an injective function. If the color histograms of both graphs $c^{(k)}(G) = \{\!\{c_v^{(k)} : v \in V\}\!\}$ and $c^{(k)}(G') = \{\!\{c_v^{(k)} : v \in V'\}\!\}$ are different at any $k$, the graphs are not isomorphic; otherwise, if they are equal after $|V|$ iterations, the isomorphism test is inconclusive, meaning that $G$ and $G'$ may or may not be isomorphic.

**Remark 1.** *Saying that GNNs are at most as expressive than WL test means that if two nodes have the same color after $k$ iterations of WL, they must also have the same embedding after $k$ GNN layers:*

$$c_u^{(k)} = c_v^{(k)} \implies \mathbf{h}_u^{(k)} = \mathbf{h}_v^{(k)}. \tag{4}$$

*The converse holds when* UPDATE$^{(k)}$, AGGREGATE$^{(k)}$ *and* READOUT *are injective functions.*

## 2.3 GENERALIZATION

**Generalization Error.** We are given a dataset $S = (x_1, ..., x_m) \sim \mathcal{D}_m$ of $m$ independent and identically distributed (i.i.d.) samples drawn from the distribution $\mathcal{D}$ over the input space $\mathcal{X}$. Each $x_i$ is associated with a binary label $y_i \in \mathcal{Y} = \{-1, 1\}$.

We consider the following hypothesis class:

$$\mathcal{F} = \{ f : \mathcal{X} \to [\text{-1}, 1] \}. \tag{5}$$

Given a loss function $\ell : (f(x), y) \mapsto [0, 1]$, which measures the prediction error between $f(x)$ and $y$, we define the empirical risk of $f$ and its true (population) risk, respectively, as:

$$\hat{L}_S(f) = \frac{1}{m} \sum_{j=1}^{m} \ell(f(x_j), y_j), \quad L(f) = \mathbb{E}_{(x,y)\sim\mathcal{D}}[\ell(f(x), y)]. \tag{6}$$

Generalization error is defined as the *difference between true and empirical risk*. It is bounded by the combination of sample size and either Rademacher Complexity or VC Dimension, as we will detail in the following definitions.

**Rademacher complexity.** Rademacher Complexity measures how much a given function class can adjust to random noise and overfit irrelevant information in the training dataset.

We define the Rademacher variables $\boldsymbol{\sigma} = (\sigma_1, \ldots, \sigma_m)$ as i.i.d. variables where $P(\sigma_i = 1) = P(\sigma_i = -1) = 0.5$. The Empirical Rademacher Complexity $\hat{\mathcal{R}}_S(\mathcal{F})$ with respect to the sample $S$, and its true (populational) counterpart $\mathcal{R}_m(\mathcal{F})$, are respectively defined as:

$$\hat{\mathcal{R}}_S(\mathcal{F}) = \mathbb{E}_{\boldsymbol{\sigma}}\left[\sup_{f \in \mathcal{F}} \frac{1}{m} \sum_{i=1}^{m} \sigma_i f(x_i)\right], \quad \mathcal{R}_m(\mathcal{F}) = \mathbb{E}_{S \sim \mathcal{D}^m}\left[\hat{\mathcal{R}}_S(\mathcal{F})\right] \tag{7}$$

The highest possible complexity value is 1, when the hypothesis class can always fit all the random noise of the Rademacher distribution with $\sigma_i f(x_i) = 1, \forall i \in \{1, 2, ..., m\}$.

Both empirical and true Rademacher can be used to bound generalization error (Mohri et al., 2018), where, for every $\delta \in (0, 1)$, with probability at least $1 - \delta$, each of the following inequalities hold:

$$L(f) \le \hat{L}_S(f) + 2\hat{\mathcal{R}}_S(\mathcal{F}) + 3\sqrt{\frac{\log \frac{2}{\delta}}{2m}}, \quad L(f) \le \hat{L}_S(f) + 2\mathcal{R}_m(\mathcal{F}) + \sqrt{\frac{\log \frac{1}{\delta}}{2m}} \tag{8}$$

**Vapnik-Chervonenkis (VC) dimension.** VC dimension provides a *combinatorial* measure of the capacity of the hypothesis class $\mathcal{F}$. Its definition relies on the concept of *shattering*. A set of points $S = (x_1, ..., x_m)$ is said to be shattered by $\mathcal{F}$ if, for any possible labeling assignment $y \in \{-1, 1\}^m$, there exists a function $f \in \mathcal{F}$ such that $f(x_i) = y_i$ for all $i$. In other words, $\mathcal{F}$ can realize all $2^m$ possible label distributions on $S$. The VC dimension of $\mathcal{F}$, denoted by $\mathrm{VC}(\mathcal{F})$, is defined as the cardinality of the largest set $S$ that can be shattered by $\mathcal{F}$:

$$\mathrm{VC}(\mathcal{F}) = \sup \{m \in \mathbb{N} : \exists S \text{ with } |S| = m \text{ such that } S \text{ is shattered by } \mathcal{F}\} \tag{9}$$

If sets of arbitrary size can be shattered by $\mathcal{F}$, then $\mathrm{VC}(\mathcal{F}) = \infty$.

VC dimension can also be used to bound generalization (Mohri et al., 2018). If we denote $\mathrm{VC}(\mathcal{F}) = d$ then, for every $\delta \in (0, 1)$, with probability at least $1 - \delta$:

$$L(f) \le \hat{L}_S(f) + \sqrt{\frac{2d \log \frac{em}{d}}{m}} + \sqrt{\frac{\log \frac{1}{\delta}}{2m}}. \tag{10}$$

Notice that $d$ appears divided by $m$ in the equation above. This means that, roughly speaking, $\mathrm{VC}(\mathcal{F})$ also represents the amount of samples needed to assure proper generalization in $\mathcal{F}$.

## 3 NEURODIVERSITY MEETS COLORS

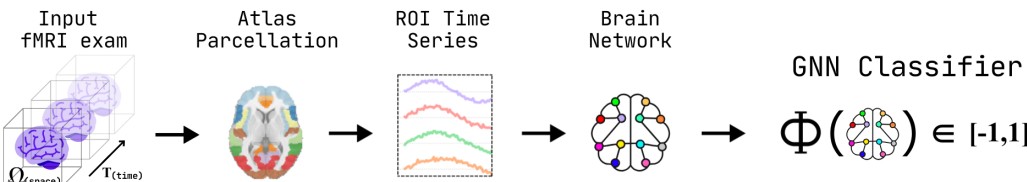

Figure 1: Illustration of the structure of the functions in our hypothesis class $\mathcal{F}$.

We start by formally defining the pipeline of our hypothesis class. Let $\mathcal{F}$ be a class of functions defined as in Equation 5. The functions in $\mathcal{F}$, also illustrated in Figure 1, have the following structure. First, the function $f \in \mathcal{F}$ maps an fMRI signal to a prediction by applying a fixed and deterministic preprocessing pipeline $\mathcal{P}$ followed by a trainable GNN $\Phi$. We can write $f = \Phi \circ \mathcal{P}$. More precisely, $f$ receives as input an fMRI signal of the form $x : (\Omega \times T) \to \mathbb{R}$, where $\Omega = \mathbb{R}^3$ represents spatial coordinates and $T = \mathbb{R}$ represents time. The map $\mathcal{P}$ converts the signal $x$ into a colored graph $G$, which will be referred to as a functional connectivity (FC) graph and will be the input of the GNN.

In what follows, we describe how an FC graph $G = \mathcal{P}(x)$ is constructed from the signal $x$. First, a brain atlas with $n$ ROIs is used to transform the signal $x$ into a multivariate time series $t = (t_1, t_2, .., t_n)$, where $t_i(\tau)$ is the average intensity of the signal in the region delimited by the i-th

ROI in $\Omega$ at time $\tau$. The ordering of the components $t_1, \ldots, t_n$ is induced by the atlas indexing and is consistent across all inputs $x \in \mathcal{X}$, and the region delimited by each ROI is the same for every $x \in \mathcal{X}$. Each ROI corresponds to a node $v \in V$ in the graph $G$, whose color $c(v)$ encodes the ROI identity (identified by a one-hot vector in an identity matrix). To define the edge set $E$, the correlation matrix $\mathbf{M}_{n \times n}$ is computed by evaluating pairwise Pearson correlations between (the time series associated with) all pairs of ROIs. Then, to obtain $E$, a thresholding function is applied to $\mathbf{M}$ to construct the binary adjacency matrix $A$, either by using a threshold value for the correlation or a percentage to select the top highest correlation values.

Lastly, the resulting graph is used as an input for a GNN $\Phi$ that outputs a value for $\Phi(G) \in [\text{-}1, 1]$. The GNN is the only trainable component of $\mathcal{F}$; the preprocessing map $\mathcal{P}$ is fixed and shared by all functions in the class. When necessary, we will introduce more restrictions to specify which functions with that structure will be contained in $\mathcal{F}$. A more detailed description of how this pipeline is carried out in practice, along with simple visualizations using real datasets, can be found in Appendix B.2.

### 3.1   SHALLOW NETWORKS

In this subsection, we restrict our analysis to "Shallow-Max GNNs", which will be defined below.

**Definition 1** (Shallow-Max GNN). *A Shallow-Max GNN is defined as a GNN that has a single message-passing layer where the neighborhood of every $u \in V$ is aggregated as:*

$$\text{AGGREGATE}^{(k)}(u) = \text{MAX}\left( \{\{\mathbf{h}_v^{(k)} : v \in \mathcal{N}(u)\}\} \right) \tag{11}$$

*where $k = 0$, and after calculating* UPDATE$^{(k)}$ *as in Equation 1, it applies a readout of the form:*

$$\text{READOUT} = \text{MAX}\left( \{\{\mathbf{h}_v^{(L)} : v \in V\}\} \right) \tag{12}$$

*where* MAX *denotes element-wise maximum and $L = 1$ is the number of layers.*

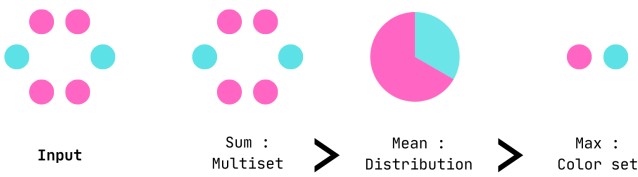

Figure 2: Hierarchy of neighborhood aggregation functions with respect to their expressivity. Sum aggregation can count the multiplicity of colors in a multiset (as in 1-WL), mean aggregation can capture their distribution, and the max aggregation used in Shallow-Max GNNs can identify the set of colors present, but cannot distinguish multiplicity. Adapted from Xu et al. (2018).

Max aggregation is less expressive than sum or mean, as illustrated in Figure 2. Since the expressive power of a GNN depends on both the aggregation/readout functions and the number of layers, Shallow-Max GNNs are at most as expressive as max aggregation. By establishing our results on this baseline, we ensure they apply broadly to architectures with equal or greater expressivity, such as GIN, as well as to commonly used, less expressive baselines like GCN (Kipf & Welling, 2016) and GraphSAGE (Hamilton et al., 2017). Moreover, using shallow models will make our results more robust to practical limitations such as over-smoothing (Li et al., 2018) and over-squashing (Alon & Yahav), which restrict the viability of using GNNs with multiple stacked layers. In fact, as we show in the following theorem, Shallow-Max are enough to achieve perfect expressivity over FC graphs, which could lead to poor generalization.

**Theorem 1** (Shallow network theorem). *Let $S = (x_1, ..., x_m)$ a sample of $m$ fMRI signals, which are mapped by $\mathcal{P}$ into $p \leq m$ different (non-isomorphic) FC graphs. If $\mathcal{F}$ contains all possible Shallow-Max GNNs, the empirical Rademacher complexity will be bounded by:*

$$\frac{2p}{m} - 1 \leq \hat{\mathcal{R}}_S(\mathcal{F}) \leq \sqrt{\frac{p}{m}} \tag{13}$$

We prove the theorem in Appendix C.1. If our hypothesis class $\mathcal{F}$ has all possible Shallow-Max GNNs, the empirical Rademacher complexity attains its maximum value in most practical scenarios, where $p = m$. The only exception is the case where FC graphs appear repeatedly, which is theoretically possible (although very unlikely) if two fMRI exams are similar enough in the input space $\mathcal{X}$. However, even if that happens, if the number of repetitions is small, then the empirical Rademacher complexity would still be close to 1.

For a more exact account of how generalization could be bounded in that scenario, without making assumptions on the data distribution, we introduce the following result. This theorem explains how position awareness enables the model to shatter a permutation orbit into $n!$ distinct colored graphs. Intuitively, this phenomenon arises because explicit ROI identification removes graph symmetries. Once every node has a unique identity, the graph admits no non-trivial automorphisms (as defined in the preliminaries).

**Theorem 2** (Shattered Orbit Theorem). *Let $S_{\mathcal{F}}$ be the number of possible non-colored FC graphs under the preprocessing pipeline $\mathcal{F}$, out of which $A_{\mathcal{F}}$ are asymmetric, and let $\Sigma_{\mathcal{F}}$ be the number of possible colored non-asymmetric graphs under the same pipeline, and $n$ be the size of the atlas. The VC dimension of the hypothesis class $\mathcal{F}$, if it contains all possible Shallow-Max GNNs, is given by:*

$$VC(\mathcal{F}) = A_{\mathcal{F}} \cdot n! + \Sigma_{\mathcal{F}}. \tag{14}$$

*Moreover, $VC(\mathcal{F})$ is bounded by $A_{\mathcal{F}} \cdot n! \leq VC(\mathcal{F}) \leq S_{\mathcal{F}} \cdot n!$, where, for large $n$, $VC(\mathcal{F})$ can be approximated by either $A_{\mathcal{F}} \cdot n!$ or $S_{\mathcal{F}} \cdot n!$, and $A_{\mathcal{F}} \cdot n!$ yields a tighter approximation.*

The theorem is proved in Appendix C.2. In fact, $VC(\mathcal{F})$ is the maximum number of non-isomorphic FC graphs in a fixed pipeline for $\mathcal{F}$, as Shallow-Max GNNs can achieve perfect expressivity over such graphs. Explicit ROI identification not only makes each FC graph immediately identifiable to 1-WL, which would not be the case if all node colors were equal, but also practically "shatters" each permutation symmetry orbit into $n!$ different equivalence classes; that shattering explicitly only affects asymmetric graph structures, but, as we note above, $VC(\mathcal{F}) \approx S_{\mathcal{F}} \cdot n!$, so the resulting effect is practically equivalent to multiplying the number of possible graphs by $n!$. In our application domain, $n$ tends to be relatively large (in the order of 100 (Cui et al., 2022)), so the VC dimension of the model will always be very large. Relating this result to the previous theorem, notice that $VC(\mathcal{F})$ is also very sensitive to atlas size, so larger atlases lead to a larger number of possible FC graphs, which can decrease the chance of FC graph repeating in each sample and slightly increase the Rademacher complexity values in Equation 13 (assuming that we do not increase sample size accordingly).

While the results in this subsection specifically address the Shallow-Max GNN to establish a baseline for perfect expressivity, they represent a lower bound on the generalization capabilities of more sophisticated models, such as deeper models or with more expressive aggregate/readout functions.

## 3.2 EXPRESSIVITY, GENERALIZATION AND NETWORK PARAMETERS

We now extend our analysis to a more standard hypothesis class $\mathcal{F}$, in which the GNN component is characterized by sum-aggregation and piece-wise linear activation functions across $L$ layers. This allows us to derive tighter generalization bounds in terms of preprocessing hyperparameters, such as atlas size and embedding dimensionality. In what follows, we show that $VC(\mathcal{F})$ can be bounded by atlas size.

**Lemma 1** (Atlas Lemma). *Suppose that the GNN of our hypothesis class $\mathcal{F}$ has a piece-wise linear activation function with two pieces, with $L$ message-passing layers where all layers have width at most equal to the atlas size $n$, their* AGGREGATE *function is a summation and its* UPDATE *function is a single-layer perceptron. Under these restrictions, the VC dimension of $\mathcal{F}$ will be bounded by:*

$$VC(\mathcal{F}) \leq O(L^2 n^2 log(nL)) \tag{15}$$

We derived this result from Morris et al. (2023). Proof is in Appendix C.3.1. This result highlights a fundamental tradeoff in brain graph learning: increasing spatial resolution through finer atlas parcellation comes at the cost of worse generalization bounds. Notice that this upper bound can be arbitrarily increased without hitting the upper bound given by Equation 14, as increasing atlas size simultaneously increases the number of possible FC graphs in $O(n!)$.

We now introduce the embedding-atlas ratio and present the following result.

**Lemma 2** (Embedding-atlas Lemma). *Suppose that our deterministic preprocessing pipeline $\mathcal{P}$ assigns to each ROI a fixed embedding of size $d$ as its feature vector, which is the same for every graph. We define an embedding-atlas ratio measure $\rho = d/n$. Suppose, additionally, that the GNN of our hypothesis class $\mathcal{F}$ has a piece-wise linear activation function with two pieces, with $L$ message-passing layers where all layers have width at most equal to $d$, their* AGGREGATE *function is a summation and its* UPDATE *function is a single-layer perceptron. Under these restrictions, the VC dimension of $\mathcal{F}$ will be bounded by:*

$$VC(\mathcal{F}) \leq O(L^2 n^2 \rho^2 log(\rho n L)) \tag{16}$$

Notice that this also holds if we use the one-hot ROI identity as an embedding, where $\rho = 1$. Therefore, the Atlas Lemma is a particular case of the Embedding-Atlas Lemma.

Proof is in Appendix C.3.2. This result suggests a quantitative tradeoff between position awareness and generalization: under a fixed atlas size, if we want tighter generalization bounds compared to the atlas lemma, we can compress positional dimensionality with $\rho < 1$, but that reduces the linear separability of the ROI embeddings; on the other hand, increasing $\rho$, while increasing linear separability, can scale VC dimension quadratically, where we can arbitrarily increase the bias of the model as a universal approximator until we eventually hit the $O(n!)$ upper bound given by the orbit shattering. This tradeoff can matter if position awareness is used as an inductive bias or a source of explainability (Zheng et al., 2024b; Li et al., 2021).

### 3.3 EXTENSION TO CONNECTION PROFILE AND ITS VARIANTS

We focused our analysis on graphs where one-hot ROI identity is the only node feature, but it is important to extend the analysis to connection profile: although ROI identity is effective at modeling positional information in a discretized and explicit manner, which is effective for our analysis, many modern models use variants of connection profile, where either the adjacency matrix (Cui et al., 2022; Kan et al., 2022) or the correlation matrix (Li et al., 2021; Zheng et al., 2024b; Xu et al., 2025a) can be used as a feature matrix. The experiments made by Cui et al. (2022) also showed that connection profile generally achieves better performance compared to other node feature choices.

In fact, much of our results can be adapted to this setting. Firstly, we show in Appendix C.3.1 that the Atlas Lemma bounds are the same for connection profile if we have the same constraints on the GNN architecture, and similarly, the Embedding-atlas Lemma bounds also hold for any $d$-dimensional node feature space given by the deterministic graph construction pipeline $\mathcal{P}$. Examples include compressing the connection profile matrix through eigendecomposition of the original matrix or selecting an arbitrary subset of its columns. Moreover, weighted connection profile can also be used to identify each ROI through the only perfect correlation value in the vector (between the given ROI and itself), as verified experimentally by Kan et al. (2022); the only rare exception would be the case where two or more distinct ROIs have perfect correlation, where they would have the exact same connection profile and become symmetric to each other. Also, notice that the single message-passing layer of the Shallow-Max GNN aggregates the identities of the neighbors of each ROI, which is practically equivalent to creating a latent representation of binarized connection profile. Therefore, weighted connection profile contains all positional information that can be perfectly expressed by Shallow-Max GNNs, while *additionally* leveraging correlation values as continuous features; that increased feature diversity may actually improve generalization by augmenting the node-level representations learned by the GNN (D'Inverno et al., 2025).

## 4    LIMITATIONS AND FUTURE WORK

We believe that future research should conduct experimental assessments on the practical implications of our theoretical findings. GNN architectures, feature construction methods and atlas size are all aspects that could be explored under empirical evaluations, as well as aspects not accounted for by our theory such as regularization hyperparameters and prior domain knowledge on neuroscience. We furthermore believe that the assumption of a fixed distribution over the data should be disputed, especially given that out-of-distribution generalization is a known challenge of the field and comes by the nature of the data collection process (Zheng et al., 2024b; Xu et al., 2025a). Moreover, the framework could be improved by a thorough connection profile analysis – which we started in Section 3.3 – expanding the results beyond certain specific assumptions (like using unweighted edges and piece-wise linear activation functions) and extending our parameter analysis to Rademacher complexity, to understand how data distribution influences our results; towards the latter objective, the generalization bounds derived by Garg et al. (2020) can be highly relevant, as they are explicitly related to permutation invariance. Lastly, given that our analysis showed that it is very easy to achieve perfect expressivity under a purely binary color-based perspective, future studies should focus on more fine-grained notions of expressivity, for example, as in Maskey et al. (2026). We expand on other promising research avenues in Appendix D.

## 5    CONCLUSIONS

In this paper, we mathematically analyze the risks that position awareness can pose for generalization in brain network analysis. We showed that position awareness allows GNNs to have perfect expressivity with a single layer, where Rademacher complexity would be the worst possible and VC dimension would explode due to node-permutation orbit shattering. We then relate our results to generalization bounds given by network parameters: we derive a VC dimension bound that is $O(n^2 log(n))$ with respect to atlas size $n$, and then relate it to the dimensionality compression given by the embedding-atlas ratio, suggesting a quantitative tradeoff between position awareness and generalization. To the best of our knowledge, this is the first paper to attempt to explain overfitting in fMRI graph learning by using classical generalization theory, and the first to provide a broad theoretical analysis of expressivity in the brain graph learning field. Our theoretical framework offers valuable insight while avoiding many common unrealistic assumptions, such as being able to stack several message-passing layers. We expect this paper to encourage a deeper theoretical discussion on the causes of overfitting in the field, guiding the future development of practical solutions to improve generalization.

## 6    AUTHOR CONTRIBUTIONS

All authors contributed to this paper. Our contributions[2], are as following:

- **Matheo Angelo Pereira Dantas:** Conceptualization, Investigation, Formal analysis, Visualization, Writing (original draft, review and editing).
- **Caterina Graziani:** Formal analysis (Appendix B.1, Proposition 1), Visualization (Figure 2), Writing (review and editing).
- **Leo Sampaio Ferraz Ribeiro:** Supervision, Writing (review and editing).
- **André Carlos Ponce de Leon Ferreira de Carvalho:** Supervision; contributed to Conceptualization and Writing (review and editing).

## 7    ACKNOWLEDGEMENTS

Matheo Angelo Pereira Dantas and André Carlos Ponce de Leon Ferreira de Carvalho were supported by the São Paulo Research Foundation (FAPESP) under the scholarship grant 24/09181-2, titled "Explainability in Graph Neural Networks for Autism Assessment Using fMRI Analysis". Caterina Graziani was supported by Project DEEP-GRAPH, funded by the Italian Ministry of University and Research (MUR) PRIN 2022 (project code: 2022YLRBTT_003, CUP: B53C24006570006).

---

[2]Based on the CRediT contributer role taxonomy: `https://credit.niso.org/`

## 8 ETHICS STATEMENT

In this section, we briefly discuss ethical considerations on our work.

### 8.1 POTENTIAL SOCIETAL HARMS

A societal harm that can result from research in brain network analysis is discrimination against its target demographics, particularly through the legitimization of the medical model of disability (Wang et al., 2025). Studies on AI-assisted diagnosis of neurological conditions often conflate disability with disease (Keyes, 2020), treat people with disabilities as a burden to society (Spiel et al., 2020), or are motivated by the objective of promoting behavioral interventions to uphold neurotypical social norms (Spiel et al., 2022). Therefore, a substantial body of current AI ethics research (Spiel et al., 2022; Maye & Hansen, 2025; Wang et al., 2025) advocates for a paradigm shift — for example, by using the neurodiversity paradigm as an alternative (Pellicano & den Houting, 2022) — and the adoption of participatory approaches (Nicolaidis et al., 2019), which can help ethically align the objectives of research with the needs of its primary stakeholders (Fletcher-Watson et al., 2019) and allow scientific investigation of phenomena that would otherwise go unnoticed without their direct perspective (Raymaker et al., 2020).

### 8.2 POSITIONALITY

Out of the four authors of this paper, three (Matheo, Leo and André) are late-diagnosed Brazilian autistic adults, with Leo additionally being diagnosed with ADHD. Our life experiences and worldviews are diverse, but from that positionality, all three of us are motivated to contribute to facilitating the diagnosis of individuals with autism and improving their mental health outcomes. This positionality also implies that our contact with neuroatypical conditions other than our own is narrower, and their demands with respect to AI-assisted diagnosis might be different from what we would expect from our own point of view. Moreover, although our approach is mostly participatory due to our own positionality, future efforts should be made towards also enabling participatory input from stakeholders who are not academics (Nicolaidis et al., 2019).

### 8.3 LLM USE DISCLOSURE

This paper does not rely significantly on content generated by Large Language Models (LLMs). LLMs were used to support text revision, literature search, and visualization (Figures 3 and 5 were partially developed with LLM-generated code).

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

APPENDIX

In the Appendix section, we include more details on related work, definitions and mathematical proof for all the theorems presented throughout the paper, as well as a "roadmap of future research directions" section with research problems that can follow up our theoretical analysis.

## A    RELATED WORK

**Position-aware features in brain graph learning.** Tang et al. (2025) provide a comprehensive list of feature engineering methods to create feature matrices in fMRI graphs, with varied degrees of position awareness[3]. Three of them are position-aware: identity matrix, where each node receives a one-hot vector (analogous to using a randomly initialized embedding given by the first GNN layer) indicating its corresponding ROI; connection profile, where the adjacency matrix is used as a feature matrix; and eigendecomposition, where a low-dimensional version of the connection profile is constructed from the eigenvectors of the original matrix. Connection profile is also often implicitly implemented as directly using the correlation matrix as a feature matrix (Zheng et al., 2024b; Li et al., 2021; Xu et al., 2025a). Three other kinds of features are not position-aware: node degree (and permutation-invariant statistics of the degrees in the neighborhood of each node), PageRank (Page et al., 1999), and DeepWalk embedding (Perozzi et al., 2014). Cui et al. (2022) benchmarked most of these methods (except for DeepWalk and PageRank) with k-fold cross-validation and showed that node identity consistently achieved bad performance (average AUC below 60% in all datasets), while connectivity profile, despite always having the highest average AUC, also had more performance instability on smaller datasets.

In addition to node feature initialization, other important components of the GNN can be position-aware. More notably, BrainGNN (Li et al., 2021) has position-aware message passing, modeling the brain as a heterogeneous graph where each ROI represents a "node type" and then using different weights for each ROI as in R-GCN (Schlichtkrull et al., 2018); also, explainability is frequently position-aware, focusing on creating class-level explanations by observing the most relevant ROIs (Li et al., 2021) and pairs of ROIs (Zheng et al., 2024b;a) for each target label, with the goal of assuring transparency in medical decision-making and studying possible neurological biomarkers.

**Generalization and overfitting in brain graph learning.** GNNs struggle with overfitting in fMRI data, where powerful models considered to be state-of-the-art, such as BrainGNN (Li et al., 2021), commonly perform below or just slightly above simple baselines, such as SVM (Xu et al., 2025a; Cui et al., 2022; Zheng et al., 2024a). Although the generalization bounds derived from classical generalization theory have been studied in a few papers on Deep Learning for neuroscience (Wang et al., 2024b; Huang et al., 2024), they have not been examined in the specific context of graph neural networks. The most probable cause of overfitting, according to modern brain network analysis research, is the scarcity of medical data (Tang et al., 2025; Luo et al., 2024)), which also amplifies the curse of dimensionality resulting from the complexity of fMRI data (Zhang et al., 2023). To address this problem, Tang et al. (2025) point that the most recent solutions rely on data augmentation (Liu et al., 2023b), generative models to create artificial samples (Zong et al., 2024), and meta-learning to "learn to learn" from multiple training tasks and datasets (Yang et al., 2022). Another issue faced in the generalization of GNNs in fMRI data is the sensitivity to distribution shifts, especially between different data collection sites (Xu et al., 2025a); to address that problem, recent models propose techniques from out-of-distribution (OOD) and domain generalization (Xu et al., 2025a; Qiu et al., 2024), as well as causal inference and information bottlenecks (Zheng et al., 2024a;b).

**Expressivity in brain graph learning.** Because node features are not natural to brain networks and have to be engineered from the ROI BOLD series, the model mostly relies on the topology of the brain graph (Cui et al., 2022), justifying the need for a theoretical analysis of expressivity. While the current literature of brain network analysis still does not have broad theoretical analyses of expressivity, multiple recent studies use the Graph Isomorphism Network architecture (Xu et al., 2018), an architecture capable of achieving the same expressive power as the 1-WL test, often with the explicit intent of achieving higher expressivity (Kim & Ye, 2020; Kim et al., 2021; Zheng et al., 2024b; Thapaliya et al., 2025; Meng et al., 2025). Other approaches propose the use of Weisfeiler-Lehman

---

[3]They don't use the same definitions of position awareness as we do; in their paper, positional features focuses on the relative position of nodes, and structural features focus on the structure of the graph.

graph kernels (Shervashidze et al., 2011) as a manual feature engineering method for classical machine learning models such as SVM (Choi et al., 2022; Li et al., 2022; Ma et al., 2023; Wang et al., 2024a).

**GNNs, expressivity and generalization.** Generalization Theory is a widely studied area in Graph Machine Learning (Vasileiou et al., 2025), and generalization is closely linked to expressivity, inasmuch as expressivity increases the number of distinguishable graph pairs and thus enhances the richness of the hypothesis class. Among various other works, Morris et al. (2023) was the first to directly relate generalization to expressivity, linking VC dimension to the number of graphs that can be distinguished by the WL test and the number of parameters of the GNN, under the restriction of using piecewise polynomial activation functions (e.g. ReLU). D'Inverno et al. (2025) extended the analysis to the case of GNNs with Pfaffian activation functions (e.g. sigmoid and hyperbolic tangent). Other works focus on Rademacher complexity as a data-dependent measure: Garg et al. (2020) derived tight generalization bounds based on the parameters of the model and its expressive power, also relating the results explicitly to permutation invariance; later, Carrasco et al. (2025) studied how the complexity can be influenced by the equivalence classes induced by the coloring algorithm (not necessarily limited to message-passing GNNs), where complexity is bounded by the number of equivalence classes in which the GNN partitions the sample and it is Lipschitz-continuous under discrete perturbations over the sample. For more detailed and up-to-date literature reviews on the topic, the reader may refer to Vasileiou et al. (2025) and Carrasco et al. (2025).

## B    DEFINITIONS AND THEORETICAL REFERENCES

### B.1    SYMMETRY GROUP

Symmetry groups play a central role in graph machine learning, since graphs exhibit discrete symmetries such as for example permutations of their nodes. These groups provide the mathematical framework for transformations that allow nodes to be arbitrarily reordered, a property that is fundamental to graph neural networks and the message-passing paradigm.

In order to keep the work self-contained, we recall in this section useful definitions from group theory and ground them in the context of graph learning. The material is largely based on Sáez de Ocáriz Borde & Bronstein (2025) and Bronstein et al. (2021).

**Definition 2.** *A **group** $\mathbb{G} = (A, \circ)$ consists of a set $A$ equipped with a binary operation $\circ : A \times A \to A$ satisfying the following axioms:*

- *Associativity: $(g \circ h) \circ k = g \circ (h \circ k)$ for all $g, h, k \in A$;*

- *Identity: there exists a unique $e \in A$ satisfying $e \circ g = g \circ e = g$ for all $g \in A$;*

- *Inverse: For each $g \in A$ there is a unique inverse $g^{-1} \in A$ such that $g \circ g^{-1} = g^{-1} \circ g = e$.*

Note that groups in general are not required to be commutative, that is we can have $g \circ h \neq h \circ g$. Moreover, the codomain of the operation $\circ$ is $A$, namely the set $A$ is closed under $\circ$.

The **symmetry group** $\mathbb{S}_n$ is the group of all node permutations $\pi : V \to V$, where $|V| = n$, and the group operation is function composition.[4] To provide an idea on how a group acts on (geometric) data we need to define the so-called *group action* and in particular we define the action of a group $\mathbb{G}$ on the space of graphs $\mathcal{G}$.

**Definition 3.** *Let $\mathbb{G}$ be a group and let $\mathcal{G}$ be the set of finite graphs. A **group action** of $\mathbb{G}$ on $\mathcal{G}$ is a mapping $\alpha : \mathbb{G} \times \mathcal{G} \to \mathcal{G}$ satisfying:*

- *Identity: $\alpha(e, G) = G$;*

- *Composition: $\alpha(g \circ h, G) = \alpha(g, \alpha(h, G))$,*

*for all $g, h \in \mathbb{G}$, $G \in \mathcal{G}$, and where $e \in \mathbb{G}$ is the identity.*

---

[4]An alternative representation for $\mathbb{S}_n$ is as the group of $(n \times n)-$permutation matrices, with matrix multiplication as the group operation.

In the main text, group actions are expressed in matrix form: a permutation $\pi \in \mathbb{S}_n$ is represented by its permutation matrix $\boldsymbol{\Pi}$, acting on node features $\mathbf{X}$ and adjacency matrices $\mathbf{A}$ as $\mathbf{X} \rightarrow \boldsymbol{\Pi}\mathbf{X}$ and $\mathbf{A} \rightarrow \boldsymbol{\Pi}\mathbf{A}\boldsymbol{\Pi}^{\intercal}$. This provides a concrete realization of the abstract group action defined above.

Last, we define the *orbit* of an element $G \in \mathcal{G}$ under the action of $\mathbb{G}$:

$$Orb(G) = \{\alpha(g, G) \,:\, g \in \mathbb{G}\}.$$

That is, the set of all graphs in which $G$ can be transformed under the action of group $\mathbb{G}$.

### B.2 THE BRAIN GRAPH LEARNING PIPELINE

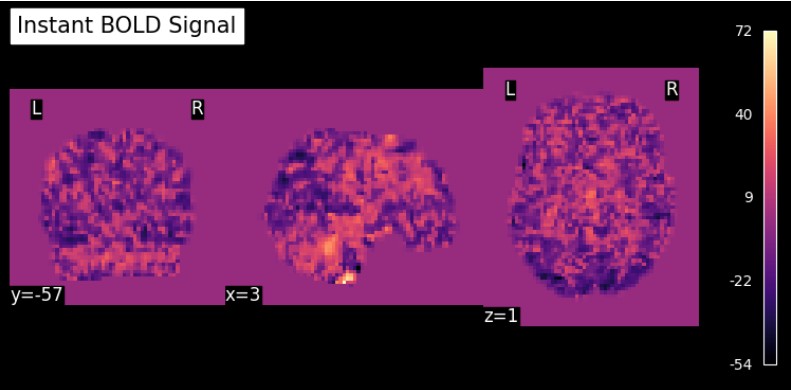

Figure 3: fMRI images of a sample sourced from ABIDE-PCP (Autism Brain Imaging Data Exchange — Preprocessed Connectomes Project) (Di Martino et al., 2014; Craddock et al., 2013). The 3D image was extracted from a specific timeframe, and for each axis in (x,y,z), the figure shows a two-dimensional slice where pixel intensity expresses the intensity of the BOLD signal.

In brain graph learning, referring more specifically to models that perform graph-level classification on static fMRI graphs using GNNs, the input of the model is an fMRI (functional Magnetic Resonance Imaging) exam, a three-dimensional exam which films brain activity (Zhang & Chiang-shan, 2012). It does so by recording the BOLD (Blood-Oxigen-Level-Dependent) signal, which indicates brain activity by mapping blood flow in each part of the brain. More specifically, the fMRI exam is a sequence of three-dimensional images, where the brightness intensity of each voxel (that is, a "3D pixel") in the image indicates the intensity of the BOLD signal within the corresponding spatial coordinate. Although our mathematical formalization assumes an input space with all possible signals in $\mathbb{R}^4$, raw fMRI exams are typically preprocessed with techniques from computational neuroscience – for example, to standardize values and remove noise (Craddock et al., 2013; Cui et al., 2022). An example of a preprocessed fMRI visualization is shown in Figure 3.

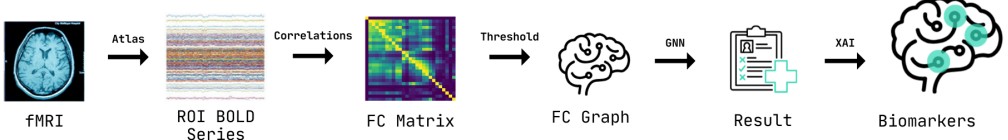

Figure 4: Illustration of the fMRI graph learning pipeline

The overall pipeline (Cui et al., 2022) is illustrated in Figure 4. Firstly, over the fMRI exam, we apply a brain atlas, a mathematical model that partitions an fMRI exam into several brain regions, called Regions Of Interest (ROI), and creates a BOLD series for each region by averaging the BOLD

the signal within each ROI. An illustration of a region partitioning produced by an atlas is showed in Figure 5. We then construct a graph, where each node corresponds to a ROI, and edges are inserted for pairs of ROIs that have a highly positive correlation between their respective BOLD series. The edges can be binarized (Luo et al., 2024; Tang et al., 2025) or weighted by the value of the correlation (Cui et al., 2022; Bessadok et al., 2023). The most used correlation function is Pearson correlation (Cui et al., 2022; Bessadok et al., 2023; Tang et al., 2025). We can select the pairs by applying a threshold to the correlation value or by using a fixed percentile of the correlation matrix. Node features are also created from the ROI multivariate time series, and the specific features vary from model to model (see the first paragraph in Appendix A). The graph is then used as input for a GNN, which outputs a binary label indicating whether the model predicts that the person has a target neurological condition such as autism. Modern models also apply techniques of Explainable AI (XAI) to view patterns in the brain that influenced the decision of the model, usually in the form of important ROIs or important ROI connections. The XAI step is important to assure transparency in medical decision-making and to support biomarker discovery (Li et al., 2021; Zheng et al., 2024b; Xu et al., 2025a).

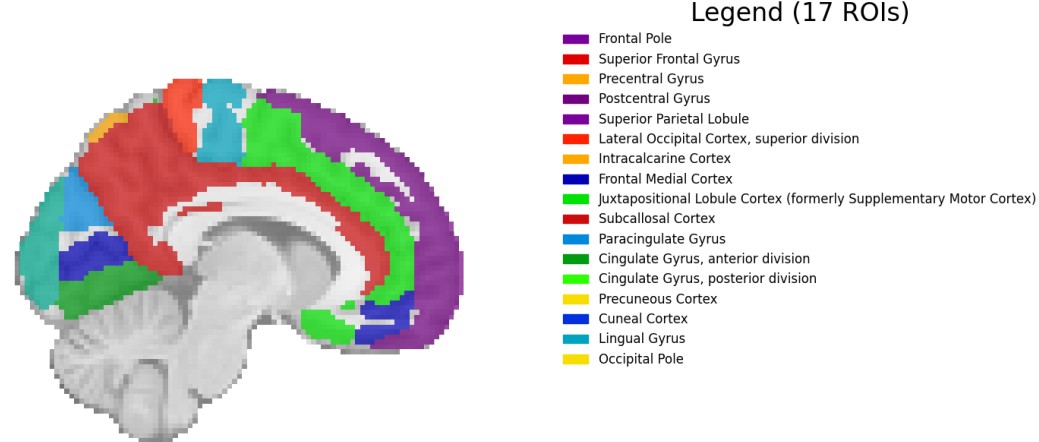

Figure 5: Illustration of a ROI division given by a Harvard-Oxford atlas parcellation from the Nilearn Python library[6] ('`cort-maxprob-thr25-2mm`') in a 2D sagittal view of the brain. Each color indicates the region delimited by the respective ROI and the corresponding legend shows its name.

## C PROOFS

Before presenting the proofs, we prove that the hypothesis class $\mathcal{F}$ defined on fMRI signals as in Section 3 and the hypothesis class $\mathcal{F}_{\text{FC}}$ defined on functional connectivity (FC) graphs, are equivalent. This observation justifies the direct adaptation of results from the graph-learning literature to the signal-level setting.

**Proposition 1** (Equivalence between signal- and graph-level hypothesis classes). *Let*

$$\mathcal{F} := \{\, f : \mathcal{X} \to [-1, 1]\,\}$$

*be the hypothesis class defined in Section 3, where each function is of the form $f = \Phi \circ \mathcal{P}$. Here, $\Phi$ denotes a GNN and $\mathcal{P} : \mathcal{X} \to \mathcal{G}_{\text{FC}}$ is a fixed preprocessing map from signals to FC graphs. Let*

$$\mathcal{F}_{\text{FC}} = \{\Phi : \mathcal{G}_{\text{FC}} \to [-1, 1]\}$$

*be the corresponding hypothesis class defined on FC graphs (namely, the class composed only of the GNN component of the pipeline). The following holds:*

---

[6]`https://nilearn.github.io/dev/modules/generated/nilearn.datasets.`
`fetch_atlas_harvard_oxford.html`

*1) for all $S = (x_1, \cdots, x_m)$,*

$$\widehat{\mathcal{R}}_S(\mathcal{F}) = \widehat{\mathcal{R}}_{\mathcal{P}(S)}(\mathcal{F}_{\mathrm{FC}}) \tag{17}$$

*where $\mathcal{P}(S) = (\mathcal{P}(x_1), \cdots, \mathcal{P}(x_m))$ are the FC graphs obtained after preprocessing;*

*2)*

$$\mathrm{VC}(\mathcal{F}) = \mathrm{VC}(\mathcal{F}_{\mathrm{FC}}). \tag{18}$$

*Proof.* We start by proving the first statement: $\widehat{\mathcal{R}}_S(\mathcal{F}) = \widehat{\mathcal{R}}_{\Phi(S)}(\mathcal{F}_{\mathrm{FC}})$. By definition of $\widehat{\mathcal{R}}_S(\mathcal{F})$ and because the functions $f \in \mathcal{F}$ are composition of a fixed preprocdessing and a trainable GNN follows that:

$$\widehat{\mathcal{R}}_S(\mathcal{F}) = \mathbb{E}_{\boldsymbol{\sigma}}\left[\sup_{f \in \mathcal{F}} \frac{1}{m} \sum_{i=1}^m \sigma_i f(x_i)\right] = \mathbb{E}_{\boldsymbol{\sigma}}\left[\sup_{\Phi \in \mathcal{F}_{\mathrm{FC}}} \frac{1}{m} \sum_{i=1}^m \sigma_i \Phi(\mathcal{P}(x_i))\right] = \widehat{\mathcal{R}}_{\mathcal{P}(S)}(\mathcal{F}_{\mathrm{FC}}).$$

Hence, given that the preprocessing is fixed for all functions $f \in \mathcal{F}$, then $\widehat{\mathcal{R}}_S(\mathcal{F})$ and $\widehat{\mathcal{R}}_{\mathcal{P}(S)}(\mathcal{F}_{\mathrm{FC}})$ are equivalent. Note that, this is true for any fixed and deterministic preprocessing map $\mathcal{P}$. If $\mathcal{P}$ is not injective, the sequence $(\mathcal{P}(x_1), \cdots, \mathcal{P}(x_m))$ possibly contains multiple copies of the same FC graph.

We now prove Eq. 18 stating the equivalence between $\mathrm{VC}(\mathcal{F})$ and $\mathrm{VC}(\mathcal{F}_{\mathrm{FC}})$. We begin by showing that $\mathrm{VC}(\mathcal{F}_{\mathrm{FC}}) \leq \mathrm{VC}(\mathcal{F})$. Assume that the hypothesis class $\mathcal{F}_{\mathrm{FC}}$ can shatter a set of FC graphs $\{G_1, \cdots, G_m\}$. Given that $\mathcal{P}$ is surjective, meaning that for every FC graph there exists (at least) one signal from which it is constructed, i.e., $\forall G_i \in \mathcal{G}_{\mathrm{FC}} \exists x_i \in \mathcal{X}$ s.t. $G_i = \mathcal{P}(x_i)$, it follows that the corresponding set of signals $\{x_1, \cdots, x_m\}$ can be shattered by $\mathcal{F}$. This implies $\mathrm{VC}(\mathcal{F}_{\mathrm{FC}}) \leq \mathrm{VC}(\mathcal{F})$. Vice versa, suppose that the set $\{x_1, \cdots, x_m\}$ is shattered by $\mathcal{F}$. This means that the GNN component of $\mathcal{F}$ must be able to distinguish the FC graphs obtained after preprocessing the signals $\{x_1, \cdots, x_m\}$. This requires that the preprocessing map assigns different FC graphs to each $x_i$, yielding a set $\{G_1, \cdots, G_m\}$ shattered by $\mathcal{F}_{\mathrm{FC}}$. Hence, $\mathrm{VC}(\mathcal{F}) \leq \mathrm{VC}(\mathcal{F}_{\mathrm{FC}})$, completing the proof. $\qquad\square$

**Remark 2.** *In what follows, we state all results at the signal level, which is the domain of primary interest in the present investigation, while deriving the proofs on the FC-graph domain. This is justified by the equivalence between $\mathcal{F}$ and $\mathcal{F}_{\mathrm{FC}}$ established in Proposition 1.*

## C.1 SHALLOW NETWORK THEOREM

In order to prove the Shallow Network Theorem (Theorem 1) we need to first state and prove the following preliminary results:

**Lemma 3** (Shallow Test Lemma). *Let $G$ and $G'$ be two graphs with $n$ nodes. We assign $n$ distinct colors to the nodes of both graphs. If the two graphs are not isomorphic, a single iteration of the 1-WL test will distinguish them.*

*Proof.* Suppose that an edge between the two nodes with colors $c_1$ and $c_2$ respectively, exists in $G$, but not in $G'$. If $v$ and $v'$ are, respectively, the nodes with the color $c_1$ in $G$ and $G'$, then:

$$c_v^{(1)} = \mathrm{HASH}(c_1, \{\{c_2, \dots\}\}) \neq c_{v'}^{(1)} \tag{19}$$

because $c_2$ is in the neighborhood color multiset for $v$, but not for $v'$. Therefore, after the first 1-WL hash, $G$ will have the color $c_v^{(1)}$ and $G'$ won't because the HASH is injective. Hence the test determines that $G$ and $G'$ are not isomorphic.

If the colors $c_1$ and $c_2$ are not connected in $G$ but are connected in $G'$, 1-WL also determines $G$ and $G'$ are not isomorphic, following the analogous argument from $G'$ to $G$. Therefore, if 1-WL cannot distinguish $G$ and $G'$, the colors $c_1$ and $c_2$ are connected in $G$ if and only if they are connected in $G'$ for any pair of colors $c_1, c_2$ out of the colors that were distributed over the $n$ nodes of both graphs.

If $V$ and $V'$ are the sets of nodes of the colored graphs $G$ and $G'$ respectively, only one mapping $f : V \to V'$ can be an isomorphism, which is the bijection that maps each node of $V$ to the

corresponding node with the same color in $V'$. The mapping $f$ will be an isomorphism because, besides preserving node colors, it also preserves all edges, as the existence of an edge between two nodes is only determined by the colors of the two nodes, in $G$ and $G'$. Therefore, 1-WL cannot distinguish $G$ and $G'$ only if they are indeed isomorphic to each other.

$\square$

**Proposition 2.** *Shallow-Max GNNs can achieve the same expressivity as 1-WL over FC graphs, thereby being perfectly expressive.*

*Proof.* Consider FC graphs constructed as in Section 3. Each node corresponds to a brain ROI and is initialized with a unique identifier (e.g., a one-hot vector), yielding an injective initial coloring of the nodes. Since the ROI indexing is fixed across inputs, the same set of node colors is shared across different FC graphs. Under this injective initialization, one iteration of the WL-test will be able to distinguish any pair of non isomorphic FC graphs, as shown in Lemma 3.

We claim that a Shallow-Max GNN can achieve the same expressivity as 1-WL on FC graphs. If the AGGREGATE, UPDATE, and READOUT functions are injective over the set of node colors, then the resulting graph representation is injective with respect to the input graph. In general, max-aggregation is not injective over multisets because it cannot distinguish two multisets with different color multiplicities (See Figure 2). However, since FC graphs are initialized with one-hot encodings as node colors, multiplicities never arise in this setting. Moreover, we can achieve the same expressive power as 1-WL by using an injective update function that generates a different one-hot vector for each possible pair of central ROI node and ROI neighborhood vector present in our sample.

$\square$

**Corollary 1.** *Suppose that the hypothesis class $\mathcal{F}$ contains all possible Shallow-Max GNNs. If the fMRI samples in the dataset $S = (x_1, ..., x_m)$ all correspond to non isomorphic FC graphs, the empirical Rademacher Complexity $\hat{\mathcal{R}}_S(\mathcal{F})$ is equal to 1.*

*Proof.* From Proposition 2 we know that Shallow-Max GNNs can achieve perfect expressivity over FC graphs. Then, we can arbitrarily pick a Shallow GNN with injective UPDATE and its readout output $h^{(L)}$ will be injective in function of the input FC graph. In our definition of $\mathcal{F}$, any function can be applied to $h^{(L)}$ to predict the binary label of the graph, so if the graphs corresponding to the samples in $S = (x_1, x_2, \ldots, x_m)$ are all non isomorphic, any possible assignment of binary labels in $\sigma : S \to \mathcal{Y}$ can be perfectly fit by $\mathcal{F}$ with $f(x_i) = \sigma_i$. In that case, the empirical Rademacher complexity $\hat{\mathcal{R}}_S(\mathcal{F})$ will be equal to 1.

$\square$

Now we are ready to prove the Shallow Network Theorem (Theorem 1):

**Theorem.** *Let $S = (x_1, ..., x_m)$ a sample of $m$ fMRI signals, which are mapped by $\mathcal{P}$ into $p \leq m$ different (non-isomorphic) FC graphs. If $\mathcal{F}$ contains all possible Shallow-Max GNNs, the empirical Rademacher complexity will be bounded by:*

$$\frac{2p}{m} - 1 \leq \hat{\mathcal{R}}_S(\mathcal{F}) \leq \sqrt{\frac{p}{m}}, \tag{20}$$

*where $p$ is the number of non-isomorphic FC graphs in the sample and $m$ is the sample size.*

*Proof.* Carrasco et al. (2025) proved that, if a given graph coloring algorithm (which may be a GNN) partitions the sample $S$ into $P$ equivalence classes, we have that:

$$\hat{\mathcal{R}}_S(\mathcal{F}) \leq \sqrt{\frac{P}{m}}. \tag{21}$$

Notice that a GNN cannot generate different outputs for two isomorphic graphs, so the number of equivalence classes cannot be greater than the number of non-isomorphic graphs in the sample. Therefore:

$$P \leq p \implies \hat{\mathcal{R}}_S(\mathcal{F}) \leq \sqrt{\frac{p}{m}} \tag{22}$$

Now we just have to prove the following lower bound:

$$\frac{2p}{m} - 1 \leq \hat{\mathcal{R}}_S(\mathcal{F}) \tag{23}$$

We will use the results proved by Carrasco et al. (2025). They showed that, given two samples of graphs $S = (G_1, \ldots, G_m)$ and $S' = (G'_1, \ldots, G'_m)$, with each graph (in both samples) associated with a color $c(G_i) = c_j \in \mathcal{GC}$, then, for every color $c_j \in \mathcal{GC}$, if the amount of graphs in both samples with color $c_j$ differs by at most $\epsilon_j$:

$$|\mu_j(S) - \mu_j(S')| \leq \epsilon_j \tag{24}$$

Then the difference between the Rademacher complexities of $S$ and $S'$ is bounded by:

$$|\hat{\mathcal{R}}_S(\mathcal{F}) - \hat{\mathcal{R}}_{S'}(\mathcal{F})| \leq \frac{1}{m} \sum_{c_j \in \mathcal{GC}} \epsilon_j. \tag{25}$$

Suppose that we have a sample of FC graphs $S = (G_1, G_2, \ldots, G_m)$ where the same isomorphism class appears multiple times, i.e. there are graph colors $c_j \in \mathcal{GC}$ induced by 1-WL on $S$ such that $\mu_j(S) > 1$. For each $c_j \in \mathcal{GC}$, we choose one graph with color $c_j$ as the original graph and any remaining graphs with the same color as repeated graphs $G_k \in \mathcal{RG}$.

Assuming that at least $m$ different FC graphs are possible in $\mathcal{D}$, we create a new sample $S' = \{G'_1, \ldots, G'_m\}$ by replacing every repeated graph in $S$ with a new one. From Corollary 1, we know that $\hat{\mathcal{R}}_{S'}(\mathcal{F}) = 1$.

If we denote $|\mathcal{RG}| = r$, each element in $\mathcal{RG}$ adds 2 to the total difference of graph colors between $S$ and $S'$ (one for removing an element of $\mathcal{RG}$ from $S$ and one for adding a previously unseen graph to $S'$). Therefore:

$$\sum_{c_j \in \mathcal{GC}} |\mu_j(S) - \mu_j(S')| = \sum_{c_j \in \mathcal{GC}} \epsilon_j = 2r \tag{26}$$

$$\Rightarrow \frac{1}{m} \sum_{c_j \in \mathcal{GC}} \epsilon_j = \frac{2r}{m} \tag{27}$$

$$\Rightarrow |\hat{\mathcal{R}}_S(\mathcal{F}) - \hat{\mathcal{R}}_{S'}(\mathcal{F})| \leq \frac{1}{m} \sum_{c_j \in \mathcal{GC}} \epsilon_j = \frac{2r}{m} \tag{28}$$

Therefore, the lower bound for $\hat{\mathcal{R}}_S(\mathcal{F})$ will be $1 - 2r/m$. But if we define $p$ as the number of non-isomorphic graphs in the sample, notice that $r = m - p$ (the number of repeated graphs is the number of graphs that exceed the amount of non-isomorphic graphs), so:

$$1 - \frac{2r}{m} = 1 - \frac{2(m-p)}{m} = 1 - \left(\frac{2m}{m} - \frac{2p}{m}\right) = 1 - 2 + \frac{2p}{m} = \frac{2p}{m} - 1 \tag{29}$$

That concludes our proof of the Shallow Network Theorem.

$\square$

C.2   SHATTERED ORBIT THEOREM

The Shattered Orbit Theorem (Theorem 2) is stated as follows:

**Theorem.** *Let $S_{\mathcal{F}}$ be the number of possible non-colored FC graphs under the preprocessing pipeline of $\mathcal{F}$, out of which $A_{\mathcal{F}}$ are asymmetric, and let $\Sigma_{\mathcal{F}}$ be the number of colored non-asymmetric graphs possible under the same pipeline, and $n$ be the size of the atlas. The VC dimension of the hypothesis class $\mathcal{F}$, if it contains all possible Shallow-Max GNNs, is given by:*

$$VC(\mathcal{F}) = A_{\mathcal{F}} \cdot n! + \Sigma_{\mathcal{F}}. \tag{30}$$

*Moreover, $VC(\mathcal{F})$ is bounded by $A_{\mathcal{F}} \cdot n! \leq VC(\mathcal{F}) \leq S_{\mathcal{F}} \cdot n!$, where, for large $n$, $VC(\mathcal{F})$ can be approximated by either $A_{\mathcal{F}} \cdot n!$ or $S_{\mathcal{F}} \cdot n!$, and $A_{\mathcal{F}} \cdot n!$ yields a tighter approximation.*

*Proof.* We start by associating with every node $u_i$ a hidden color $hc(u_i)$, injectively determined by its corresponding ROI BOLD series $t_i$.

Therefore, every FC graph $G$ can be parametrized by:

$$G_{\mathcal{F}}(t) = (\Psi_{\mathcal{F}}, \mathcal{C}), \quad \mathcal{C} = \{(c(u_i), hc(u_i))\}_{i=1}^n, \tag{31}$$

where $t = (t_1, \ldots, t_n)$ is the multivariate ROI BOLD series associated with $G$, $c(u_i)$ is the ROI color associated with node $u_i$ and $\Psi_{\mathcal{F}}(hc(u), hc(v))$ is a function that calculates the correlation between the ROI BOLD series associated with $u$ and $v$, and applies a threshold to determine the existence of an edge $(u, v)$ (the threshold can be fixed for all graphs, or determined by the graph in the case we are extracting a fixed top percentile of edges).

In our definition of atlas, every ROI multivariate BOLD series $t = (t_1, \ldots, t_n)$ is realized by assigning a constant BOLD series value $t_i$ over the whole three-dimensional region within the $i-$th ROI in $\Omega$. Also, permuting the BOLD series of the ROIs does not change the thresholds in $\Psi_{\mathcal{F}}(hc(u), hc(v))$, hence we are always allowed to apply every permutation $\pi$ over the hidden colors. Under the natural action of the permutation group $S_n$ (cf. Appendix B.1), this is equivalent to keeping the hidden colors fixed and applying the inverse permutation $\pi^{-1}$ to the ROI-identifying node colors, which produces a valid FC graph. Given that Shallow-Max GNNs can achieve perfect expressivity over FC graphs, as proven in the Shallow Network Theorem, the VC dimension of $\mathcal{F}$ is equal to the maximum number of possible FC graphs under the chosen pipeline. We can separate these isomorphism classes as:

$$VC(\mathcal{F}) = \Lambda_{\mathcal{F}} + \Sigma_{\mathcal{F}} \tag{32}$$

Where $\Lambda_{\mathcal{F}}$ is the number of isomorphism classes on asymmetric graphs and $\Sigma_{\mathcal{F}}$ is the same for non-asymmetric graphs. We can further represent $\Lambda_{\mathcal{F}}$ as $A_{\mathcal{F}} \cdot n!$, where $A_{\mathcal{F}}$ is the number of structural isomorphism classes of asymmetric graphs under the chosen pipeline. It will be multiplied by $n!$ because, following our definitions of automorphism and asymmetric graphs, permuting the colors of an asymmetric FC graph, which always leads to a valid FC graph, also always leads to a new non-isomorphic graph, given that the colors are all different. Therefore, if the resulting FC graph is asymmetric, its underlying graph structure admits only the trivial structural automorphism. Therefore every permutation of the hidden colors produces a graph that is not structurally isomorphic to the others, yielding $n!$ distinct FC graphs. If the FC graph admits non-trivial structural automorphisms, some permutations produce isomorphic graphs. These graphs contribute to the correction term $\Sigma_F$. Summing the contributions of all possible FC graphs gives:

$$VC(\mathcal{F}) = A_{\mathcal{F}} \cdot n! + \Sigma_{\mathcal{F}}. \tag{33}$$

Because $\Sigma_{\mathcal{F}} > 0$, we have that $A_{\mathcal{F}} \cdot n! \leq VC(\mathcal{F})$ is a lower bound for $VC(\mathcal{F})$. Also, if we denote $Z_{\mathcal{F}} = S_{\mathcal{F}} - A_{\mathcal{F}}$ as the number of possible non-colored non-asymmetric FC graphs in $\mathcal{F}$, we have that $Z_{\mathcal{F}} \cdot n! \geq \Sigma_{\mathcal{F}}$ because each non-asymmetric FC graph has $n$ nodes where $n$ colors will be distributed, so for each non-colored FC graph, the number of ways of coloring it is at most equal to $n!$; with that, we stablish that $S_{\mathcal{F}} \cdot n!$ is an upper bound for $VC(\mathcal{F})$:

$$VC(\mathcal{F}) = A_{\mathcal{F}} \cdot n! + \Sigma_{\mathcal{F}} \leq A_{\mathcal{F}} \cdot n! + Z_{\mathcal{F}} \cdot n! \tag{34}$$

$$= (A_{\mathcal{F}} + Z_{\mathcal{F}}) \cdot n! \tag{35}$$

$$= (A_{\mathcal{F}} + (S_{\mathcal{F}} - A_{\mathcal{F}})) \cdot n! \tag{36}$$

$$= S_{\mathcal{F}} \cdot n! \tag{37}$$

$$\implies VC(\mathcal{F}) \leq S_{\mathcal{F}} \cdot n! \tag{38}$$

Therefore, we have that $A_{\mathcal{F}} \cdot n! \leq VC(\mathcal{F}) \leq S_{\mathcal{F}} \cdot n!$. Moreover, for $n \to \infty$ we have that $S_{\mathcal{F}} \to A_{\mathcal{F}}$ because the proportion of non-asymmetric graphs tends to 0 (Bollobás, 2001). Hence, for large $n$, $VC(\mathcal{F}) = O(A_{\mathcal{F}} \cdot n!)$.

However, $A_{\mathcal{F}} n!$ yields a tighter approximation. When we distribute the $n$ colors over the nodes, each non-colored graph can produce $n!/s_G$ colored graphs, where $s_G$ is the number of distinct isomorphism-preserving permutations of colors (which corresponds to the number of structural automorphisms) in the graph. In non-asymmetric graphs, $s_G \geq 2$ (as it has the trivial automorphism plus at least one non-trivial automorphism), where, if we had $s_G = 2$ for every non-symmetric graph, $VC(\mathcal{F})$ would be halfway between $A_{\mathcal{F}} \cdot n!$ and $S_{\mathcal{F}} \cdot n!$; however, we can trivially create graphs where $s_G > 2$: for example, we can generate an arbitrary BOLD series and attribute it to $k > 2$ ROIs, leading to a clique of size $k > 2$ where all nodes have the same connection profile and neighbors; in that case, all of these nodes can be permuted in automorphisms of $G$, where we have $s_G \geq k!$. Therefore:

$$|A_{\mathcal{F}} \cdot n! - VC(\mathcal{F})| < |S_{\mathcal{F}} \cdot n! - VC(\mathcal{F})|. \tag{39}$$

That concludes our proof of the Shattered Orbit Theorem.

$\square$

### C.3 VC DIMENSION AND GNN PARAMETERS

In order to prove Lemma 1 and Lemma 2, we introduce the necessary definitions given by Morris et al. (2023).

**Definition 4** (Color Complexity)**.** *The **color complexity** of a graph $G$, denoted by $u$, is defined as the number of distinct colors computed by the 1-dimensional Weisfeiler-Lehman algorithm (1-WL) after $|V(G)|$ iterations on $G$.*

**Definition 5** (Graph Class $\mathcal{G}_{d,\leq u}$)**.** *Let $\mathcal{G}_{d,\leq u}$ denote the class of all graphs endowed with $d$-dimensional vertex features (in $\mathbb{R}^d$) that have a color complexity of at most $u$.*

We consider a specific class of GNNs, denoted as $GNN_{slp}(d, L)$, where the aggregation function is a summation the update and readout functions are implemented as Single Layer Perceptrons (SLP) with width at most equal to d. We can now state the following result regarding the VC dimension for the case where the activation functions are piece-wise linear (degree $\delta = 1$).

**Proposition 3** (VC Dimension of $GNN_{slp}$ with Piece-wise Linear Activations)**.** *Let $d, L \in \mathbb{N}$ and consider the class $\mathcal{F} = GNN_{slp}(d, L)$ equipped with **piece-wise linear** activation functions consisting of $q$ pieces (polynomial degree $\delta = 1$).*

*Let $\Theta = d(2dL + L + 1) + 1$ be the number of parameters. For any color complexity bound $u \in \mathbb{N}$, the VC dimension of $\mathcal{F}$ on the graph class $\mathcal{G}_{d,\leq u}$ is bounded by:*

$$VC_{\mathcal{G}_{d,\leq u}}(\mathcal{F}) \leq \mathcal{O}(L \cdot \Theta \log(q \cdot u \cdot \Theta)). \tag{40}$$

We will suppose that $q = 2$ (as in ReLU or LeakyReLU) to simplify the expression. First, $u$ is at most equal to $n$, as the color complexity cannot exceed the size of the graph and the size of an FC graph is always equal to atlas size $n$. Next, we simplify $\Theta = d(2dL + L + 1) + 1$ as $O(d^2L)$, so $L\Theta = O(d^2L^2)$. With that in mind, VC dimension can be bounded by:

$$VC_{\mathcal{G}_{d,\leq n}}(\mathcal{F}) \leq \mathcal{O}(d^2 L^2 \log(2 \cdot n \cdot (d^2 L))) = \mathcal{O}(d^2 L^2 \log(ndL)) \tag{41}$$

### C.3.1    ATLAS LEMMA

For the Atlas Lemma (Lemma 1), because we are using an $n-$dimensional one-hot vector as the feature vector of each node, we have $d = n$. Therefore, VC dimension is bounded by:

$$VC_{\mathcal{G}_{n,\leq n}}(\mathcal{F}) \leq \mathcal{O}(n^2 L^2 \log(n \cdot n \cdot L)) = \mathcal{O}(n^2 L^2 \log(nL)) \tag{42}$$

Notice that this also holds if we use an $n-$dimensional connectivity profile vector instead of a one-hot identity vector, as we would have $d = n$ and color complexity will still be at most equal to $n$.

### C.3.2    EMBEDDING-ATLAS LEMMA

For the Embedding-Atlas Lemma (Lemma 2), where the size of the ROI embeddings is arbitrary, we define $\rho = d/n$, so $d$ can be also expressed as $\rho \cdot n$ in Equation 41. Therefore, given that the embeddings are fixed and not trainable, $VC(\mathcal{F})$ is bounded by:

$$VC_{\mathcal{G}_{d,\leq n}}(\mathcal{F}) \leq \mathcal{O}((\rho n)^2 L^2 \log(n \cdot (\rho n) \cdot L)) = \mathcal{O}(\rho^2 n^2 L^2 \log(\rho nL)) \tag{43}$$

Similar to the previous case, this holds for any deterministic feature construction method with $d$ dimensions, given by the preprocessing pipeline $\mathcal{P}$. This could involve applying any fixed function to connection profile, such as using an arbitrary subset of columns of the connection profile matrix or an eigendecomposition (Cui et al., 2022) of the same matrix.

## D    ADDITIONAL PROMISING RESEARCH PROBLEMS

Here, we present a list of research problems that seem promising as extensions of our findings. First, we highlight questions within the field where our work could support the search for answers.

- **Graph coarsening in brain network analysis.** Graph coarsening, also known as pooling, is a common technique used to regularize the learning process by hierarchically simplifying the representation of the graph while preserving its topology (Limbeck et al., 2025; Li et al., 2021). This is especially relevant given that we bounded the VC dimension with $O(n^2 log(n))$ with respect to the size of the input atlas. Since pooling operations are applied after the first message-passing layer, where Shallow-Max GNNs with unbounded width can already attain maximal expressivity, hierarchical pooling may help mitigate generalization issues.

- **Position awareness and explainability.** From the embedding-atlas lemma, we saw that compressing positional information can tighten the generalization bounds of the Atlas Lemma. However, this could lead to a decrease in explainability, as the model will have more difficulty in identifying specific relevant regions in the brain that become more tightly clustered in the embedding space. Future studies could employ metrics like fidelity and sparsity (Yuan et al., 2022) to assess instance-level explainability, and other metrics designed specifically to evaluate biomarker discovery from class-level explainability (Girish et al., 2024). This could be evaluated either by doing experiments or by establishing direct mathematical relations with the embedding-atlas ratio.

- **Balancing expressivity and generalization.** Despite the strong tradeoff with generalization, expressivity may be useful in fMRI analysis (Liu & Zhang, 2025), where the connectivity patterns are complex, so future works can explore means of leveraging the expressive power of position-aware GNNs while mantaining safe generalization guarantees. A possible idea to exploit is the training color diversity induced by using connection profile as the node feature vector, which may improve generalization, as mentioned in Section 3.3.

- **Comparing GNNs with other deep learning approaches.** While GNNs focus on the connectivity structure of data, other kinds of deep learning models have different geometric priors, such as CNNs (Kawahara et al., 2017; Lin et al., 2022) and LSTMs (Dvornek et al., 2017). Future studies could evaluate how effective these architectures are when dealing

with positional information in fMRI data, compared to GNNs, where we basically have to remove its core geometric priors and introduce generalization risks.

- **Deep Sets and connection profile:** given how our analysis showed that shallow-max GNNs are perfectly expressive over FC graphs, and connection profile contains all positional information that can be perfectly expressed by Shallow-Max GNNs, future studies could explore the use of max-aggregation Deep Sets (Soelch et al., 2019) as universal approximators for connection profile matrices, through theoretical analysis and experiments.

Next, we list paths in which the scope of our work could be expanded beyond binary graph-level brain network classification with message-passing GNNs.

- **Going beyond binary classification.** Our results are specific to binary graph-level classification, which is compatible with the current state-of-the-art in generalization theory for GNNs Vasileiou et al. (2025). However, the medical field may involve more complex tasks, such as disease stage prediction (Scheltens et al., 2021), brain age estimation (Pina et al., 2022) and decision-making optimization (Ferrer, 2025). Future developments could extend our analysis to such tasks.

- **Exploring architectural variations.** There are graph learning models in the literature that process graphs without traditional GNN graph-level classifiers, such as graph transformers (Kan et al., 2022), hypergraph neural networks (Liu et al., 2023a), spatio-temporal GNNs (Kim et al., 2021), and models that perform node-level classification on populational graphs (Wang et al., 2022); our theory cannot be directly applied to those cases and would need adaptation.

- **Multi-atlas models.** Another possible direction is to investigate atlas size bounds in multi-atlas models. While our results are based on a static atlas and bound generalization with their size, there are studies in the literature proposing a multi-atlas approach, processing functional connectivity across different parcellations and granularity levels (Lee et al., 2024; Xu et al., 2025b; Zhang et al., 2024a).

- **Going beyond fMRI data.** Although fMRI is the most common type of brain network data in GNN applications, other kinds of data and graph construction methods can be used, such as Diffusion Tensor Imaging (DTI), Diffusion Spectrum Imaging (DSI), electroencephalography (EEG), and morphological graphs (Luo et al., 2024; Bessadok et al., 2023). Future work may adapt our results to these kinds of exams.

