# OpenReview forum: "Neurodiversity Meets Colors: Does Position Awareness Destroy Generalization in Brain Graph Learning?"
_ICLR.cc/2026/Workshop/GRaM — ICLR 2026 Workshop GRaM Poster_

### Official Review · Reviewer_1K5y · 2026-02-15
**The work provides key problem analysis and mathematical justification, but lacks experiments.**

**Rating:** 7
**Confidence:** 2

**Review:**

The authors of this paper investigate the reasons for the weak generalization of current GNN-based AI techniques in the context of fMRI image analysis, focusing on the limitations of using ROIs. I believe that this work can provide guidance for future research in this field and, with its rigorous mathematical analysis, serve as a theoretical foundation for subsequent studies. The main shortcoming, however, is the lack of experimental comparisons or visualizations, which would have more comprehensively demonstrated the authors’ points.

1. The mathematical sections of this paper form the main body; it may be helpful to appropriately add some visualizations related to fMRI and GNNs, or simple demonstrations using specific datasets, to aid in understanding the overall problem.

**Pmlr Suitability:**

Yes

---

### Official Review · Reviewer_3te5 · 2026-02-22

**Rating:** 7
**Confidence:** 4

**Review:**

This is a very strong theoretical paper showing that adding position awareness to brain graphs harms the Rademacher complexity and will vastly increase the VC dimension. There are some issues with presentation that weaken the paper and make it harder to read, as well as some typos. I have listed specific comments below.

Comments:
- The name symmetric graph usually means a graph where the automorphism group acts transitively, which is a stronger condition than in the paper.
- The fact that position awareness corresponds to graphs with no automorphisms was not readily apparent for someone with no brain graph experience, so it was hard to connect the theorems of the paper to the main thesis. Adding an explanation would be very helpful.
- In theorem 1 it should be that the preprocessing step maps the samples into p distinct graphs not that F does.

**Pmlr Suitability:**

Yes

---

### Official Review · Reviewer_Vh8P · 2026-02-24

**Rating:** 6
**Confidence:** 2

**Review:**

**Strengths:** This paper theoretically demonstrates that position aware node features in brain network gnn can make the hypothesis class highly expressive even with shallow message passing, which in turn yields worst-case capacity bounds. Additionally, it derives VC bounds and a tradeoff via an embedding-atlas ratio.

**Weaknesses:** Nevertheless, as the paper explicitly notes, it lacks experiments although the theory’s soundness and practical predictive value would benefit such evidences.

**Workshop fit:** The paper seems a good fit.

**Pmlr Suitability:**

Yes

---

### Meta-Review · Area_Chair_2ZPD · 2026-02-25

**Decision:**

Accept

**Metareview:**

This paper covers a relevant topic to GRaM, addressed with an original idea and meaningful theoretical results. The reviewers are in agreement to accept. The authors should address the reviewers' suggestions concerning clarity in the final version, and consider adding simple demonstrations on a real dataset.

**Relevance To Proceedings:**

Yes — suitable for PMLR (long paper)

**Relevance To Workshop:**

Yes — suitable for GRaM

---

### Decision · Program_Chairs · 2026-03-02

Accept (Poster)